# Optimization of Multi-Factor Model in Quantitative Trading Based On Reinforcement Learning

**Xiaotong LIN**
Division of Financial Technology
The Chinese University of Hong Kong
Shatin, Hong Kong
1155154800@link.cuhk.edu.hk

**Ming ZHANG**
Division of Financial Technology
The Chinese University of Hong Kong
Shatin, Hong Kong
1155150138@link.cuhk.edu.hk

## Abstract

Quantitative trading strategies play an important role in stock trading, and reinforcement learning (RL) has been increasingly applied to trading activities in recent years. In this paper, we mainly study the optimization of multi-factor model in quantitative trading by using the method of RL, that we train an agent with a series of historical trading data of the stock market. From the experiment results, we can see that RL is feasible in solving and optimizing similar investment decision-making problems in financial field, which can help us to obtain more stable returns. Eventually, we hope that our simple work can make more and more people notice the application of RL in investment.

Please refer to the link for the video: `https://cuhk.zoom.us/rec/share/GvBjlaPRhm6Vq_3di6CgW_Mk7h1jPe4SoETtZkJr5d26gOkTphrydKrW-_cOFdDI.ZsueNMu7WAzmatNH`

**Keywords:** Quantitative trading, Reinforcement learning, Multi-factor model

## 1 Introduction

Quantitative trading (6) consists of trading strategies that rely on mathematical models, which can avoid some irrational investments caused by subjective judgments of people. With the help of those profitable trading strategies, investors can optimize capital allocation and maximize investment performance. Trading strategy can be mainly divided into Stock (25), Macro (8), as well as Arbitrage Strategy (7), and in particular, the stock strategy (25) mainly includes Alpha (4), Commodity Trading Advisor (CTA) (27) and High Frequency Strategy (23). In this paper, we mainly focus on the Alpha Strategy.

Alpha (31) is a measure of non-systemic risk, which is used to measure the active return on an investment. So Alpha Strategy (4) is to obtain alpha returns that exceed the market index while avoiding systemic risks, which includes the multi-factor strategy (2) that is widely used in it. It establishes a model by selecting a series of factors that are most related to the rate of return, so as to give corresponding trading signals to investors. And the biggest advantage of it is that there are always some factors that can play a role in different markets and market conditions.

In recent years, due to a series of breakthroughs brought by deep convolutional neural networks, deep learning techniques attract many attentions in various fields, including in finance. For example, (29; 15) use deep learning models for financial prediction and classification, which embed a large collection of information into investment decisions. However, most financial data are time-varying, that is to say, they are unstable and noisy, so sometimes deep learning models can not achieve good performance in financial market.

On the other hand, reinforcement learning (RL) (18; 26) has made remarkable achievements in recent years, which is widely used in various scenarios, including robotics (20), game playing (21) as well

as finance (28), especially the use of deep reinforcement learning (DRL) to solve financial problems. Compared with supervised deep learning prediction model, DRL uses a reward function to optimize future rewards and it does not need large labeled training datasets, which makes it very suitable for financial trading due to the frequent changes in financial markets (32).

Actually in a multi-factor model, inappropriate weights of factors will usually lead to a bad prediction result. So in this paper, we are going to optimize the combination of factors in a strategy by DRL. The rest of this paper is structured as follows. In Section 2, we briefly review some backgrounds about alpha strategies and RL, which are related to our work. In Section 3, we describe the framework of our network. In Section 4, we shows the experiment results and analyse them. Finally we draw conclusions in Section 5.

## 2   Related Work

We begin by briefly reviewing some backgrounds about alpha factors, which is pertinent to our work. In 1993, Fama proposed the three factor model (14) and later Carhart published the four factor model (5) in 1997. In recent years, researchers and dealers have discovered many factors, which were published by WorldQuant (19). Based on the factors, many researchers have developed their trading strategies to invest in the financial markets. However, those strategies are calculated only based on stock returns, which means that they are unable to combine other relevant factors together well.

In the context of the applications of reinforcement learning (RL) using in quantitative trading, Zhang *et al.* proposed a trend reversion model based on RL for stock trading rule discovery called eTrendRev, which can produce higher returns with lower risks and recognize significant market turning points (34). And Wang *et al.* integrated deep attention networks with a RL framework to achieve a risk-return balanced investment AlphaStock strategy, which performs pretty good in U.S. and Chinese stock markets (30).

In addition, recent applications of deep reinforcement learning (DRL) in financial markets focus on discrete or continuous state and action spaces (32), such as critic-only approach (13), which is the most frequent application of DRL in financial markets. For example, it uses Deep Q-learning (DQN) and its improvements to train an agent on a single stock or asset (9; 11; 16). Besides, Yang *et al.* Used DRL to establish a robust and stable model for automated stock trading, which tested on the 30 Dow Jones stocks and outperformed the three individual algorithms and two baselines in terms of the risk-adjusted return measured by the Sharpe ratio (33).

The actor-only approach is the second most common approach. It has been used in cryptocurrency portfolio management (17), financial signal representation and trading (12).The key advantage of this approach is the continuous action space of the agent and the typically faster convergence of the learning process.The actor-critic approach is the third category and aims at combining the advantages of the actor-only and the critic-only approach. It has been applied in trading strategies (3) (22) (35).The key idea is the actor learns to choose the action which is considered best by the critic and the critic learns to improve its judgment.

From the related work mentioned above, we can see that DRL is able to solve complex sequential decision-making problems (24), which is effective in quantitative trading. So in this paper, we would to like to discuss the feasibility and effect of DRL method in improving the traditional quantitative trading strategy, that is, to further optimize the combination of factors in a strategy by using DRL method.

## 3   Our Work

### 3.1   Alpha Strategy

Alpha strategy is mainly designed to obtain excess returns, it means the strategy performs better than the index. Alpha strategy is usually a multi-factor strategy, and factors are generally generated by basic financial data, quantity data and price data. In this kind of strategy, we need to find effective factors in the market, and these factors can point out buying and selling signals of stocks. By using these factors, we can construct an alpha strategy.

In our multi-factor model, rate of return decomposition is the basic assumption. We believe that the stock is exposed to a variety of different risk factors at the same time, and the joint action among them forms the fluctuation of stock price. Therefore, we are committed to finding common factors that have an impact on most stock price fluctuations, which are called style factors, and this part of returns are called style returns. Besides, the unexplained part of style factor is considered to be the unique attribute of individual stock, which is called idiosyncratic factor, and this part of return is naturally called alpha return. So the stock return can be described as:

$$Stock\ Return = Style\ Return + Alpha\ Return \tag{1}$$

The characteristic of style income is significant but unpredictable, while that of alpha is significant and predictable to some extent. In our model, we select 20 alpha factors from 101 Formulaic Alphas (19), which can be divided into five categories: price, volume, binary classification, bias and volatility.

## 3.2 Structure Design

The framework of our work is shown in the figure below:

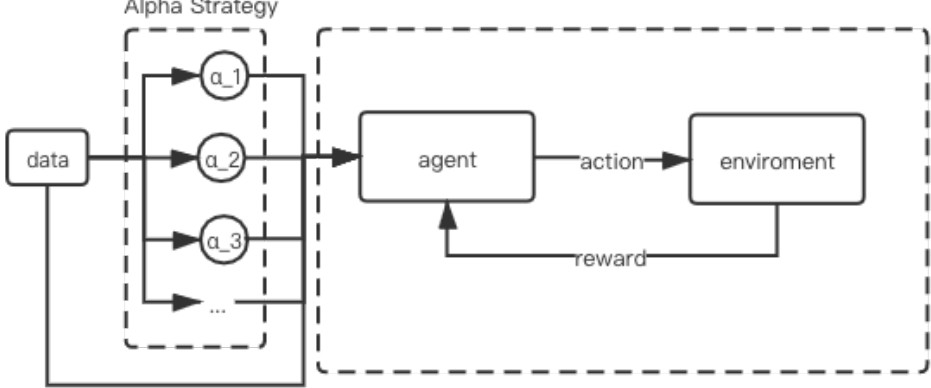

Figure 1: Framework of our work

As shown in the figure1, we firstly take the trading data of the same period as the input of the selected alpha strategies to respectively predict the trend of the stock in the next stage. Then we feed the prediction results and the original trading data into the agent at the same time and here we use the method of deep reinforcement learning. After the agent makes action, we compare the results with the real trend of the stock in the next stage, so the agent could adjust the weight by backward propagation and achieve the best learning effect in the end. In addition, we hope that the transaction style of the agent is relatively stable, so we do not prefer to maximize the return or increase the punishment for mistakes when designing the reward function. After that, we execute the trading according to the prediction results of the agent.

We use Advantage Actor-Critic (A2C) for policy optimization. We add a baseline to the Q value so that the feedback has both positive and negative feedback, so the gradient can be described as:

$$\nabla \bar{R}_\theta = \frac{1}{N} \sum_{n-1}^{N} \sum_{t=1}^{T_n} (Q^{\pi_\theta}(s_t^n, a_t^n) - V^{\pi_\theta}(s_t^n)) \nabla \log p_\theta(a_t^n | s_t^n)$$

However, in this case, we need to have two networks to calculate the state action value Q and the state value V respectively, so we change it to:

$$Q^\pi(s_t^n, a_t^n) = E[r_t^n + V^\pi(s_{t+1}^n)]$$

$$Q^\pi(s_t^n, a_t^n) = r_t^n + V^\pi(s_{t+1}^n)]$$

At this time, the Critic becomes the network that estimates the state value V. Therefore, the loss of Critic network can be described as:

$$loss = \frac{1}{N} \sum_{n-1}^{N} \sum_{t=1}^{T_n} (r_t^n + V^\pi(s_{t+1}^n) - V^\pi(s_t^n))^2$$

## 4 Experiments

### 4.1 Data Selection and Pre-processing

We use part of the historical trading data of the component stocks of CSI 300 index (10) in China stock market, which could avoid the extreme event on the stock that will adversely affect the final results. In particular, we collect the historical daily stocks trade data from 01/06/2005 to 01/10/2020 from JoinQuant (1). The data mainly include: trading day, stock code, opening price, closing price, lowest price, highest price, average price, as well as amplitude. And we select the first 80% of transaction data in the dataset for training and verification, and the last 20% for testing.

In order to speed up the training, we first pre-process the data. In other words, we first use the alpha strategy to predict the data. We select 20 alpha factors from 101 Formulaic Alphas (19). Then we use these 20 factors to calculate the data of the previous trading day, so as to help us judge whether to buy, sell or not in the next trading day. For the prediction results and corresponding actions, we define it as:

$$action = \begin{cases} buy, result > 0, \\ no \quad action, result = 0, \\ sell, result < 0 \end{cases} \tag{2}$$

### 4.2 Network

We use the deep reinforcement learning (DRL) method to train the network, which is trained on the dataset that we mentioned above. The network structure is shown in the figures below and it takes about 3.5 hours to train on a single 1-GPU machine under this setting.

| Layer (type) | Output Shape | param |
|---|---|---|
| dense_1 (Dense) | (None, 10, 16) | 480 |
| dense_2 (Dense) | (None, 10, 32) | 544 |
| dense_3 (Dense) | (None, 10, 16) | 528 |
| flatten_1 (Flatten) | (None, 160) | 0 |
| dense_4 (Dense) | (None, 3) | 483 |

Figure 2: Actor Network

| Layer (type) | Output Shape | param |
|---|---|---|
| dense_5 (Dense) | (None, 10, 16) | 480 |
| dense_6 (Dense) | (None, 10, 32) | 544 |
| dense_7 (Dense) | (None, 10, 16) | 528 |
| flatten_2 (Flatten) | (None, 160) | 0 |
| dense_8(Dense) | (None, 1) | 161 |

Figure 3: Critic Network

## 4.3 Results and Analysis

We implement the network on the test set. The figure below shows the prediction results of the network on the trading behavior of the next trading day according to the data of the previous trading day in the statistical time:

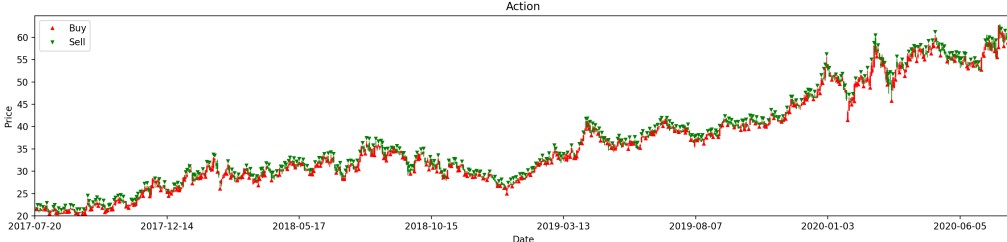

Figure 4: Trading behavior of the next trading day according to the data of the previous trading day

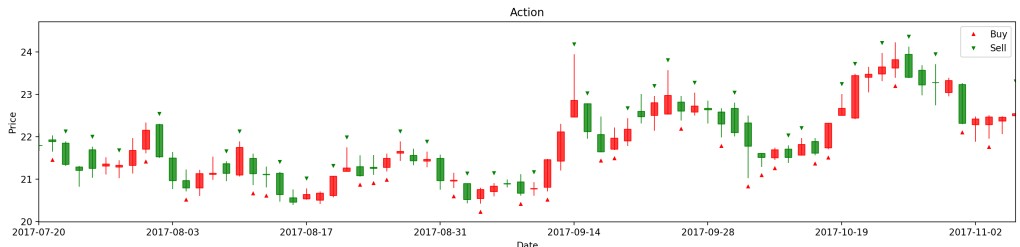

Figure 5: Trading behavior in part of the statistical time

Here we assume that we have one million principal and each transaction is a full position transaction. We calculate the return of the network after trading according to the above prediction results, and compare it with the performance and mean value of 20 alpha factors:

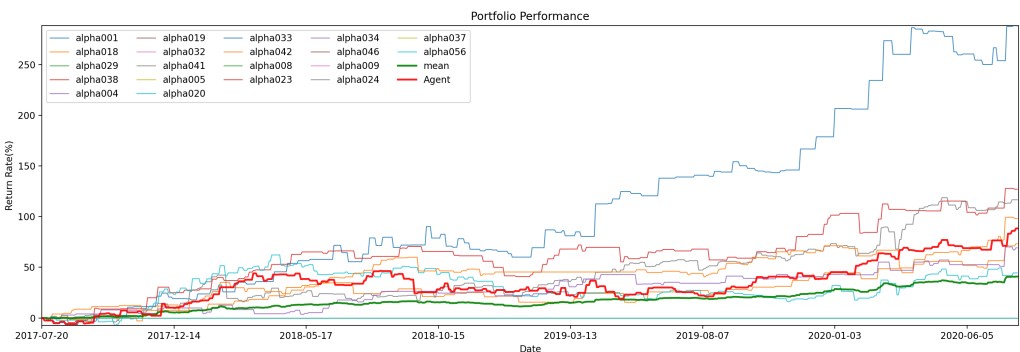

Figure 6: Comparison of the performance of our network and single traditional quantitative trading strategies

Figures 6 and 7 show that from July 2017 to Jun 2020, the strategy achieved an annualized excess return of 17.21%, an information ratio of 3.24, and a maximum withdrawal of 5.92%, which occurred between August 2017 and September 2017, and the overall return of the portfolio is pretty good. It is worth noting that in 2018, when the market style changed greatly, by the end of April, the strategy still achieved an annualized excess return of about 24%, showing that the source of strategic income and portfolio structure are relatively independent of the market style.

| Annualized excess return | 17.21% |
|---|---|
| Information ratio | 3.24 |
| Maximum drawdown | 5.92% |
| Maximum drawdown period | 2017/08-2017/09 |

Figure 7: Performance Statistics of Our Work

On the whole, from the results, we can see that some alpha trading styles are more radical, and their risks and returns are relatively large, while some alpha are more conservative and have no trading signal during the test time. In contrary, our network combines different alpha strategies, which enables trading agents to integrate their observed environment, including the stock price information and decisions of different alpha strategies, to make better trading choices.

## 5 Conclusion

In this paper, we use reinforcement learning method to improve the multi factor model of traditional quantitative trading strategy. From the results, we can see that reinforcement learning method is feasible in solving and optimizing similar investment decision-making problems in financial field. Compared with the use of single traditional quantitative trading strategy, our network can combines different alpha strategies, which enables the network to integrate their observed environment to make better trading decisions, which is more stable and can help us to obtain more stable returns.

In the follow-up work, we will explore the implementation of multiple financial assets. Besides, we find that it is hard to converge in the training of our work, so we could try to further optimize our design of network. In addition, we can get further refinements of the model in identifying the changes in trends of the market.

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
