# OpenReview forum: "Optimization of Multi-Factor Model in Quantitative Trading Based On Reinforcement Learning"
_CUHK.edu.hk/2021/Course/IERG5350_

### Official Review · AnonReviewer4 · 2020-12-18
**Very technical paper, but can be hard to understand for non financial experts**

**Rating:** 7
**Confidence:** 3

**Review:**

Overall, the author achieves good returns from the stock markets by using deep learning strategy. However, it can be elliptic to non financial experts, partly due to the technical complexity, but also due to the lack of context giving while introducing your network structure. Sentence can be clearer and more concise and informative. For example, trim those running sentence to make it semantically clearer.

Pros: Lots of technical details of trade markets, like alpha strategy and style return.Very exciting finding and good markets return. Cannot wait to see it be implement in the stock market to earn some profits.

Cons: As I mentioned, the paper lacks context while talking about the structure of your network. Because of that I cannot understand how you define your loss, what is your Q, etc. You do not define action in the structure design section while mentioning it. I know you define it later, but by switching the order it appears much clearer to readers. Also, I talk about running sentences and grammatical error above, I will just give you some examples and advice as to how to revise them.

1. So Alpha Strategy (4) is to obtain alpha returns that exceed the market index while avoiding systemic risks, which includes the multi-factor strategy (2) that is widely used in it.
Advice: This is a running sentence, try not to include four clauses in a sentence.

2. For example,
(29; 15) use deep learning models for financial prediction and classification, which embed a large collection of information into investment decisions.
Advice: Please avoid using in-line citation as subject of a sentence.

3. We add a baseline to the Q value so that the feedback has both positive and negative feedback, so the gradient can be described as:
Advice: Another running sentence. Try to avoid connecting three sentences with two conjugations.

Those are just for your reference. I hope it can help you.

---

### Official Review · AnonReviewer2 · 2020-12-20
**The overall quality of the project is good, but some improvements are needed.**

**Rating:** 7
**Confidence:** 4

**Review:**

Report summary:

This project applies deep reinforcement learning (DRL) to optimizing quantitative trading strategies, which automatically adjusts the weights of factors in a multi-factor model to pursue higher excess returns. Specifically, the authors feed the prediction results and the original trading data into the agent so it can learn the trends of the stock in the next stage. Experimental results show that the overall return of the learned portfolio is pretty good, which demonstrate DRL is feasible in solving and optimizing similar investment decision-making problems in financial field.


The paper is well-writted and -organized. The experiments show promising results, so I think it is a good example of the application of reinforcement learning. However, there are some room for improving the quality of the paper:


Major issues:

* A clear mathematical definition for different elements of RL is needed. For example, in the second equation, what exactly are $a_t^n$, $s_t^n$, and $\pi_\theta$?
* The authors mentioned that 20 factors are selected. I think they can be listed in the report, which give a better context for the problem.
* To be more reader-friendly, more background knowledge can be added. For example, a detailed introduction of alpha strategy. The authors has mentioned that "Alpha is a measure of non-systemic risk", but what is non-systemic risk? Also, what are market index, excess return, information ratio, etc.?


Minor issues:
* Missing numbers for equations.
* A typo: At line 5 of the third paragraph of Related Work: Used -> used

---

### Official Review · AnonReviewer3 · 2020-12-20
**Some comments about this paper**

**Rating:** 5
**Confidence:** 2

**Review:**

This paper proposes a reinforcement learning based quantitative trading method. But here I have three concerns:

1. The state space or environments of RL parts should be fully described in Sec.3.2, e.g. the price, or maybe the observations of selected alpha factors...

2. A2C is applied for policy optimization. This should be declare that why A2C is better that others.

3. The experiments reveal that trained rl agent can surpass the average performances of all 20 alpha factors. One problem is, since the agent is trained with different factors selections, how about changing the applied alpha factors?